# Treatment of Acute Respiratory Distress Syndrome Caused by COVID-19 with Human Umbilical Cord Mesenchymal Stem Cells

**DOI:** 10.3390/ijms24054435

**Published:** 2023-02-23

**Authors:** Tetiana Bukreieva, Hanna Svitina, Viktoriia Nikulina, Alyona Vega, Oleksii Chybisov, Iuliia Shablii, Alina Ustymenko, Petro Nemtinov, Galyna Lobyntseva, Inessa Skrypkina, Volodymyr Shablii

**Affiliations:** 1Laboratory of Biosynthesis of Nucleic Acids, Institute of Molecular Biology and Genetics, Department of Functional Genomics, National Academy of Science, 150 Zabolotnogo Str., 03143 Kyiv, Ukraine; 2Placenta Stem Cell Laboratory, Cryobank, Institute of Cell Therapy, 03035 Kyiv, Ukraine; 3Department of Infectious Diseases, Shupyk National Healthcare University of Ukraine, 04112 Kyiv, Ukraine; 4Endoscopic Unit, CNE Kyiv City Clinical Hospital # 4, 03110 Kyiv, Ukraine; 5Laboratory of Cell and Tissue Cultures, Department of Cell and Tissue Technologies, Institute of Genetic and Regenerative Medicine, State Institution, 04114 Kyiv, Ukraine; 6National Scientific Center “Institute of Cardiology, Clinical and Regenerative Medicine n.a. M. D. Strazhesko”, National Academy of Medical Sciences of Ukraine, 03680 Kyiv, Ukraine; 7Laboratory of Pathophysiology and Immunology, D. F. Chebotarev State Institute of Gerontology of the National Academy of Medical Sciences of Ukraine, 04114 Kyiv, Ukraine; 8Coordination Center for Transplantation of Organs, Tissues and Cells, Ministry of Health of Ukraine, 01021 Kyiv, Ukraine

**Keywords:** COVID-19, acute respiratory distress syndrome, mesenchymal stem cells, cytokine, microRNA, correlation, dynamic changes

## Abstract

This study aimed to identify the impact of mesenchymal stem cell transplantation on the safety and clinical outcomes of patients with severe COVID-19. This research focused on how lung functional status, miRNA, and cytokine levels changed following mesenchymal stem cell transplantation in patients with severe COVID-19 pneumonia and their correlation with fibrotic changes in the lung. This study involved 15 patients following conventional anti-viral treatment (Control group) and 13 patients after three consecutive doses of combined treatment with MSC transplantation (MCS group). ELISA was used to measure cytokine levels, real-time qPCR for miRNA expression, and lung computed tomography (CT) imaging to grade fibrosis. Data were collected on the day of patient admission (day 0) and on the 7th, 14th, and 28th days of follow-up. A lung CT assay was performed on weeks 2, 8, 24, and 48 after the beginning of hospitalization. The relationship between levels of biomarkers in peripheral blood and lung function parameters was investigated using correlation analysis. We confirmed that triple MSC transplantation in individuals with severe COVID-19 was safe and did not cause severe adverse reactions. The total score of lung CT between patients from the Control and MSC groups did not differ significantly on weeks 2, 8, and 24 after the beginning of hospitalization. However, on week 48, the CT total score was 12 times lower in patients in the MSC group (*p* ≤ 0.05) compared to the Control group. In the MSC group, this parameter gradually decreased from week 2 to week 48 of observation, whereas in the Control group, a significant drop was observed up to week 24 and remained unchanged afterward. In our study, MSC therapy improved lymphocyte recovery. The percentage of banded neutrophils in the MSC group was significantly lower in comparison with control patients on day 14. Inflammatory markers such as ESR and CRP decreased more rapidly in the MSC group in comparison to the Control group. The plasma levels of surfactant D, a marker of alveocyte type II damage, decreased after MSC transplantation for four weeks in contrast to patients in the Control group, in whom slight elevations were observed. We first showed that MSC transplantation in severe COVID-19 patients led to the elevation of the plasma levels of IP-10, MIP-1α, G-CSF, and IL-10. However, the plasma levels of inflammatory markers such as IL-6, MCP-1, and RAGE did not differ between groups. MSC transplantation had no impact on the relative expression levels of miR-146a, miR-27a, miR-126, miR-221, miR-21, miR-133, miR-92a-3p, miR-124, and miR-424. In vitro, UC-MSC exhibited an immunomodulatory impact on PBMC, increasing neutrophil activation, phagocytosis, and leukocyte movement, activating early T cell markers, and decreasing effector and senescent effector T cell maturation.

## 1. Introduction

COVID-19 has expanded internationally, resulting in an ongoing pandemic that has infected over 755 million individuals and killed over 6.8 million people in over 200 nations (https://covid19.who.int/ (accessed on 14 February 2023)). COVID-19 causes fever, fatigue, muscular discomfort, diarrhea, and pneumonia, and it can be fatal in severe cases [1].

The levels of proinflammatory cytokines rise dramatically in peripheral blood plasma during COVID-19 progression [2]. The cytokine levels of IL-2, IL-6, IL-7, G-CSF, IP10, MCP1, MIP1A, and TNF-α are determined and were mostly found to be elevated in severe COVID-19 patients with ARDS development related to lung damage and tissue fibrosis [3]. This ‘cytokine storm’ syndrome is accompanied by a decline in the overall number of T cells, T helper (CD4^+^) and cytotoxic (CD8^+^) T cells, natural killer (NK) cells, and regulatory T cells, which compromises the immune system [4]. Furthermore, patients infected with SARS-CoV-2 had high levels of monocytes and neutrophils [5,6]. Therefore, the normalization of COVID-19-caused immune system imbalance is a promising task for the effective treatment of this disease.

Nowadays, particular interest is focused on detecting circular plasma microRNA as key regulators of specific gene expression and diagnostic markers. Numerous studies indicate miR-146, miR-27, miR-221, miR-21, miR-126, miR-92a-3p, and miR-133 are dysregulated in COVID-19 [7]. miR-27a-3p was shown to inhibit M2 macrophage polarization in an acute lung injury model in mice [8]. miR424-5p is significantly upregulated in COVID-19 patients with thrombotic disease [9]. In addition, increased miR-21 expression was reported in neutrophils and macrophages in a sepsis model in mice [10]. Thus, studying the impact of the therapy with umbilical cord mesenchymal stem cells (UC-MSC) on the plasma levels of these miRNAs may help to understand their therapeutic effect on the development of acute respiratory distress syndrome (ARDS) caused by COVID-19.

MSC therapy is a promising approach to preventing the development of ARDS and multiple organ dysfunction caused by COVID-19. Based on the results of already completed clinical trials, the safety and efficacy of MSC as a regulator of the cytokine storm in treating ARDS were proven [3,11,12,13,14,15,16,17,18,19,20,21,22,23]. In addition, the potential therapeutic effects of MSC include the reduction of inflammation, prevention of tissue fibrosis, protection of alveolar cells, and stimulation of the regeneration of inflamed tissue, which can be strongly beneficial for COVID-19-injured organs [24,25,26,27,28]. The MSC transplantation does not require HLA-typing, and these cells have weak immunogenicity [29,30]. The fetoplacental complex and umbilical cord (UC) in particular are rich sources of mesenchymal stromal cells (MSC) with high proliferative properties. UC-MSC possess great immunomodulatory properties, can alter immune cell function, modulate immune responses, and reduce inflammation-induced lung injury in different pre-clinical and in vitro models [31,32]. The analysis of the clinical trials suggests that MSC fail to exert significant anti-inflammatory effects in the context of COVID-19 [33], although the anti-inflammatory actions of MSC may attenuate inflammation in the lung through different mechanisms described earlier [34,35]. Previously, MSC were shown to modulate the development, activation, and chemotaxis of dendritic cells (DCs), T cells, and B cells. MSC may be used to augment and maintain the percentage of CD28^+^ T cells in humans [36]. Thus, the study of the influence of UC-MSC on the immune cells of COVID-19 patients could be promising for recognizing their therapeutic potential in the treatment of COVID-19-caused ARDS.

However, different protocols are currently in use for treating COVID-19 with MSC therapy varying doses of injected cells, transplant rate, and intervals between MSC infusions [37]. Moreover, numerous data are collected about the impact of MSC on the levels of proinflammatory cytokines and different immune cell counts during COVID-19 development [28]. To date, the mechanism of the therapeutic effect of MSC on the course of ARDS, acute myocardial damage, and the state of the immune system in patients with COVID-19 have not yet been thoroughly studied. The levels of microRNAs in the plasma of patients with COVID-19, as key regulators of gene expression, have not yet been sufficiently studied, especially after MSC transplantation. In addition, the influence of UC-MSC in vitro on the transcriptome and secretome of peripheral blood mononuclear cells (PBMC) from patients with ARDS caused by COVID-19 as well as on the maturation of T cells has not yet been rigorously studied. Therefore, the investigation of changes in multiple biological molecules participating in pathophysiological processes will assist in identifying the main targets of MSC, expand our understanding of their therapeutic potential, and help adjust treatment protocols.

## 2. Results

### 2.1. Basic Characteristics of the Patients with COVID-19

A total of 28 patients with severe COVID-19 admitted to the Kyiv Clinical Hospital #4 were enrolled in this study after obtaining written informed consent. Detailed patient characteristics are shown in Table 1. The median ages for the MSC and Control groups were 58.0 [32–71] and 62.0 [32–73.0] years, respectively, and the intervals from illness onset to hospital admission for the MSC and Control groups were 12.3 ± 3.37 and 11.0 ± 2.69 days, respectively.

### 2.2. The Main Safety Results

The safety of the UC-MSC application was assessed using adverse events (AEs) that were recorded within 24 h after each infusion, including skin color and measurements of the patient’s blood pressure, body temperature, and pulse. Evaluation of the presence of any side effects was carried out from the moment of the first infusion until the end of the trial. The triple infusion of cells did not show significant side effects.

### 2.3. Computed Tomography of the Lung

The total lung computed tomography (CT) scores in both groups decreased gradually from all assessment periods and became significantly different on week 48 (*p* < 0.05) (Figure 1). At the same time, every point differed from another inside each group. However, for the Control group, the total score did not change in the time period from 24 to 48 weeks, when this parameter decreased for the MSC group (Appendix A).

### 2.4. The Dynamic Profile of Hematological Parameters in Patients with COVID-19

Blood parameter comparisons in patients with COVID-19, depending on the time of assessment, are presented in Figure 2. The differences in the white blood cells (WBC), granulocytes (Gra), and lymphocytes (Lym) counts as well as the neutrophil (Neu), eosinophils (Eo), and monocytes (Mo) percentages did not reach significance between the two groups, but the dynamic profile showed some fine changes. Thus, the count of WBC increased on day 7 with a subsequent decrease on day 28 in both the MSC and Control groups. In the MSC group, the count of granulocytes (Gra) decreased on day 28 compared to day 0 significantly, in contrast to the Control group. Neutrophil (Neu) percentages gradually decreased during the first 28 days in both groups, but the percentage of banded neutrophils on day 14 was significantly lower in the MSC group. A slight but significant increase in the percentage of eosinophils (Eo) on day 28 was observed in the MSC group compared to the beginning of observation. The percentage of Eo increased two-fold from days 14 to 28 in the Control group. The percentage of monocytes (Mo) did not change during the first four weeks in the Control group, whereas in the MSC group, it increased significantly from day 14. Compared to the beginning of hospitalization, the percentage of lymphocytes (Lym) increased in both groups and reached a significant difference between groups on day 28. Furthermore, the lymphocyte count doubled on day 7 in the MSC group and day 14 in the Control group, respectively. Compared to the initial day, ESR decreased in both the MSC and Control groups on days 7 and 28, respectively (Figure 2A and Appendix A). 

On the initial day, 5 out of 14 (35.71%) patients in the Control group and 2 out of 13 (15.38%) patients in the MSC group had leucopenia; 3 out of 15 (20,0%) patients in the Control group and 3 out of 13 (23.07%) patients in the MSC group had leukocytosis on day 7 (Figure 2B). Lymphopenia was observed in 12 out of 14 (85.71%) patients in the Control group and 7 out of 13 (53.84%) patients in the MSC group on day 7. On day 28, 1 out of 10 (10%) patients in the MSC group had lymphopenia, whereas no lymphopenia was observed in the Control group (Figure 2C).

### 2.5. The Dynamic Changes in Cytokine and miRNA Levels in the Plasma of Patients with COVID-19

Cytokine levels’ changes depending on the time of assessment are presented in Figure 3 and Appendix A. C-reactive protein (CRP) levels started to normalize from day 0 and day 7 in the MSC group and Control group, respectively. In the MSC group, the level of interferon gamma-induced protein 10 (IP-10) decreased on day 7 compared to the beginning of hospitalization. IP-10 was significantly higher in the MSC group compared to the Control group on days 7 and 28. The plasma level of monocyte chemoattractant protein-1 (MCP-1) slightly increased on day 7 in the MSC group, while, in the Control group, the elevation of MCP-1 concentration in patients’ plasma was observed on days 14 and 28. The content of macrophage inflammatory protein-1 alpha (MIP-1α) was significantly higher on days 7 and 28 in the MSC group compared to the Control group. The level of interleukin-10 (IL-10) was significantly higher on day 7 in the blood of patients in the MSC group in comparison to the Control group. Granulocyte colony-stimulating factor (G-CSF) content gradually decreased during the observation period in patients from the Control group in contrast to the MSC group, in which it increased consistently throughout the observation period; it reached a significant difference between groups on days 14 and 28. Interleukin 2 (IL-2) and interleukin 6 (IL-6) did not change significantly during the observation period in both groups. Plasma concentrations of IL-2, IL-6, and MCP-1 did not differ significantly in the MSC and Control groups.

The concentration of N-terminal prohormone of brain natriuretic peptide (NT-proBNP) in the plasma of MSC patients rose significantly on day 28 compared to days 0 and 14. The insignificant elevation of the plasma level of NT-proBNP on day 28 was observed in the Control group. Troponin I (TnI) did not change in MSC patients, whereas in the Control group, it decreased significantly on days 7, 14, and 28 compared to the beginning of observation.

After MSC transplantation, a significant decrease in surfactant D (SP-D) was observed in patients’ blood on days 7 and 28. Interestingly, in Control patients, there was an increase in SP-D level from the 14- to 28-day markers. Soluble form of the receptor for advanced glycation endproducts (sRAGE) was significantly reduced after 7 days in Control patients and increased on days 14 and 28. In the MSC group, there was a slight decrease in sRAGE over the entire period of hospitalization.

Changes in the relative expression levels of miRNAs, depending on the time of assessment, are presented in Figure 4 and Appendix A. In the MSC group, the expression level of miR-27a-3p was significantly different on days 7 and 28 compared to the beginning of hospitalization. miR-146a-5p, miR-21-5p, and miR-221-3p decreased on days 7 and 28 in both groups compared with the beginning of hospitalization. For example, the relative expression level for miR-21-5p and miR-146a-5p decreased more than fourfold, and for miR-221-3p, it decreased more than threefold in the Control group but only twofold in the MSC group. On day 14 of observation, the relative expression level of miR-126-3p differed significantly from day 0 in the Control group, whereas in the MSC group, it did not change for 28 days. In the MSC group, the level of miR-133a-3p decreased threefold by day 7. The plasma levels of miR-92a-3p in patients of the Control group on days 0 and 7 of hospitalization were significantly lower compared to those in the MSC group. The expression levels of miR-92a-3p and miR-424-5p were elevated in both the Control and MSC groups over the observation period.

### 2.6. Correlation between Hematological Parameters, Cytokines, and miRNA in the MSC Group

All data on the correlation analysis are available in Appendix A, and some are presented in Appendix A. CRP levels correlated negatively with the percentage of lymphocytes (r = −0.562, *p* ≤ 0.0001) and positively with ESR (r = 0.628, *p* ≤ 0.0001) and neutrophil percentage (r = 0.411, *p* ≤ 0.01). SP-D levels correlated with ESR (r = 0.386, *p* ≤ 0.01) and the concentration of IP-10 (r = 0.310, *p* ≤ 0.05). IP-10 negatively correlated with lymphocyte count (r = −0.551, *p* ≤ 0.001), eosinophils (r = −0.384, *p* ≤ 0.01), and miR-27a-3p (r = −0.472, *p* ≤ 0.005). Troponin I levels correlated with the expression of miR-133a-3p positively (r = 0.372, *p* = 0.05) and that of eosinophils negatively (r = −0.352, *p* ≤ 0.05). sRAGE plasma levels positively correlated with MCP-1 concentration (r = 0.516, *p* ≤ 0.001) and MIP-1α (r = 0.299, *p* ≤ 0.05). MCP-1 concentration correlated negatively with miR-21-5p (r = −0.396, *p* ≤ 0.01). miR-92a-3p correlated negatively with CRP (r = −0.477, *p* ≤ 0.001), SP-D (r = −0.450, *p* ≤ 0.005), and IP-10 (r = −0.409, *p* ≤ 0.005). Negative correlations were found between miR-424-5p and CRP (r = −0.435, *p* ≤ 0.005), SP-D (r = −0.452, *p* ≤ 0.005), and IP-10 (r = −0.436, *p* ≤ 0.005).

### 2.7. Correlation between Hematological Parameters, Cytokines, miRNA, and CT in the MSC Group

ESR was positively correlated with CT parameters of the lung on weeks 2 (r = 0.445, *p* ≤ 0.005), 24 (r = 0.457, *p* ≤ 0.005), and 48 (r = 0.398, *p* ≤ 0.05). The number of granulocytes positively correlated with lung fibrosis intensity on weeks 24 (r = 0.457, *p* ≤ 0.005) and 48 (r = 0.449, *p* ≤ 0.005). CRP levels positively correlated with lung damage on weeks 2 (r = 0.439, *p* ≤ 0.005), 8 (r = 0.386, *p* ≤ 0.05), 24 (r = 0.510, *p* ≤ 0.001), and 48 (r = 0.475, *p* ≤ 0.005). MCP-1 levels positively correlated with total score of CT on weeks 8 (r = 0.471, *p* ≤ 0.005) and 48 (r = 0.499, *p* ≤ 0.005). SP-D concentration correlated with CT total scores on week 2 (r = 0.373, *p* ≤ 0.05). The levels of RAGE positively correlated with CT scores on weeks 2 (r = 0.437, *p* ≤ 0.005), 8 (r = 0.436, *p* ≤ 0.01), 24 (r = 0.420, *p* ≤ 0.01), and 48 (r = 0.510, *p* ≤ 0.005).

miR-424a-5p correlated negatively with lung damage degree on weeks 2 (r = −0.571, *p* ≤ 0.001), 8 (r = −0.426, *p* ≤ 0.01), and 24 (r = −0.351, *p* ≤ 0.05). miR-21a-5p levels negatively correlated with CT total score on weeks 2 (r = −0.412, *p* ≤ 0.01), 8 (r = −0.479, *p* ≤ 0.005), and 48 (r = −0.353, *p* ≤ 0.05). During week 2, CT total scores correlated negatively with relative expression levels of miR-92a-3p (r = −0.480, *p* ≤ 0.005).

### 2.8. Effect of UC-MSC on PBMC In Vitro: Transcriptome, Flow Cytometry, and ELISA

We compared the transcriptome profiles of PBMC co-cultured with and without UC-MSC to identify alterations in gene expression levels. Four thousand one hundred seventy-two genes were differentially expressed between UC-MSC and Control samples; of these, 2993 were upregulated and 1179 were downregulated in PBMC after co-culture with UC-MSC (Figure 5A,B). Gene ontology (GO) molecular function enrichment analysis of upregulated genes revealed that serine hydrolase, serine-type peptidase, serine-type endopeptidase, endopeptidase, and metallopeptidase activities were the most significantly overrepresented (Figure 5C). GO biological process analysis showed that genes involved in an extracellular matrix organization, angiogenesis, immune defense (especially in neutrophil-mediated immunity), and leukocyte migration were upregulated. GO molecular function showed that among the downregulated genes, the most relevant categories were catalytic activity, acting on DNA, chromatin binding, and tubulin binding. GO biological process enrichment analysis of downregulated genes demonstrated that genes involved in cell division were significantly underrepresented (Figure 5D).

KEGG annotation revealed that UC-MSC induced inflammatory-associated signaling pathways (B-cell receptor and NF-kappa B signals) in PBMC. Furthermore, the enrichment of genes associated with phagosomes, cytokine–cytokine receptor interaction, protein digestions, and ferroptosis indicate the elevation of immune defense machinery in PBMC. Downregulated enriched categories analyzed by KEGG included genes associated with cell proliferation (especially DNA replication), the cell cycle, and the Fanconi anemia pathway. In addition, the apoptosis-associated genes in PBMC after co-culture with UC-MSC had decreased expression levels (Figure 5E,F).

Flow cytometry analysis demonstrated the increase in CD69^+^CD25^−^ and CD69^+^CD25^+^ cell subpopulations among CD3 T cells in PBMC after co-culture with UC-MSC. With UC-MSC, the percentage of effector T cells (CXCR3^+^), senescent effector CD8 and CD 4 T cells (CD57^+^CXCR3^+^), and memory CD8 T cells (CD57^+^CXCR3^+^CD45RO^+^) decreased in PBMC (Figure 6A). UC-MSC did not have any effect on the maturation of CD3^+^CD4^+^CD8^−^Th cells, cytotoxic CD3^+^CD4^−^CD8^+^ T cells, and regulatory T cells (CD3^+^CD4^+^CD8^−^CD25^low^CD127^low^) during co-culturing with PBMC. An ELISA of conditioned media from PBMC and UC-MSC/PBMC cultures showed the significant increase of MCP-1, IL-6, and G-CSF as well as the remarkably attenuated secretion of MIP-1α, IL-10, and IL-12p70. The concentration of IL-2 and IP-10 did not change significantly in conditioned media collected from PBMC and UC-MSC/PBMC cultures (Figure 6B). Co-culturing with UC-MSC significantly decreased the proliferation of PBMC that coincides with the downregulation of genes involved in cell division (Figure 5D and Figure 6C).

## 3. Discussion

In the present study, we aimed to reveal the impact of human umbilical cord-derived mesenchymal stem cell transplantation on the safety and clinical outcomes of patients with COVID-19 in the context of changes in lung functional status, miRNA, and cytokine levels. The positive influence of MSC therapy on lungs in both short-term and long-term periods was shown previously in severe COVID-19 patients [37]. Furthermore, the improvement of chest CT results in the first month after MSC infusion in comparison to a placebo was published by other authors [20,23]. Herein, we observed a significant reduction in lung damage during a one-year follow-up in the MSC group compared to the Control group. In MSC patients, lung CT total scores gradually decreased from week 2 to week 48 of observation, demonstrating a beneficial long-term therapeutic effect of MSC on lung lesions, whereas in the Control group, an improvement was observed only up to week 24, with residual fibrotic changes occurring by week 48. In our study, the long-term (up to one year) benefits in ARDS patients’ lungs, as seen on CT, corresponded to data concerning MSC transplantation in COPD with high systemic inflammation [38]. According to prior research, after intravenous transplantation, MSC can accumulate in the lung capillaries for a short time and can minimize the damage of alveolar epithelium and lung fibrosis [3]. In addition, MSC may have a preventive impact when applied at the beginning of illness. Although numerous preclinical studies in ARDS have demonstrated that MSC transplantation can significantly attenuate inflammation and improve the repair of damaged lung tissue, the results of our and other clinical trials are ambiguous, highlighting the need to better understand these mechanisms in patients with ARDS caused by COVID-19.

According to [39], patients with COVID-19 widely present myelocytosis and lymphopenia. However, on day 28, we found an increase in lymphocyte content after MSC therapy, which was also indicated in another study [12]. The boosted content of banded neutrophils in patients’ blood that contributed to COVID-19 severity was reported previously [40]. At the same time, a clinical trial reported that MSC transplantation significantly reduced neutrophil counts after two and four months [41]. In our study, we observed a significant decrease in the content of banded neutrophils on day 14 after the beginning of cell therapy. We supposed that an observed reduction in the range of these cells might indicate attenuation of the inflammatory process after MSC therapy.

Cytokine over-release syndrome is one of the significant factors influencing COVID-19 mortality and morbidity, and it is brought on by immune cells generating pro-inflammatory cytokines in a positive feedback cycle. Elevated blood levels of CRP, pro-inflammatory cytokines, and ESR are characteristics of a ‘cytokine storm’ [42]. MSC transplantation may function as an immune modulator in the formation of a cytokine storm brought on by inflammation, according to earlier research [43]. In this study, the level of CRP in the blood of patients with COVID-19 in the MSC group positively correlated with lung damage score based on CT, ESR, and neutrophils, and it correlated negatively with percentages of lymphocytes, which was also described earlier in patients with conventional therapy [44]. Notably, we observed a 7-day delay in CRP decline in the Control group compared to the MSC group. Thus, we hypothesize that MSC transplantation suppressed inflammation caused by COVID-19 in a paracrine manner, improving lymphocyte recovery and leading to a rapid decrease in such inflammation markers as ESR and banded neutrophils. These data are consistent with results reported by other authors [12,14,21,36,45].

Clinical studies have recommended using serum SP-D levels as a biomarker in acute and chronic respiratory diseases, such as chronic obstructive pulmonary disease and idiopathic pulmonary fibrosis. SP-D is released by type II alveolar cells, and increased serum concentrations result from protein translocation emerging with impairment of the structural integrity of the alveolar–capillary membrane [46]. The type II alveolar epithelial is the main target of SARS-CoV-2 [47]; thus, inhibition of its injury may benefit from cell therapy. Furthermore, the positive correlation of SP-D with CRP was previously published [46]. We observed a positive correlation between SP-D and lung damage score on CT, inflammatory markers of ESR, and IP-10. In addition, the plasma level of SP-D decreased on days 7 and 28 after MSC transplantation, suggesting attenuation of lung inflammation and inhibition of type II alveolar epithelium death, as was also reported in [48].

The high concentration of a lung injury marker specific for type I alveolar epithelium, sRAGE, in the blood is a prognostic biomarker of the development of ARDS and severe stages of COVID-19 [48,49,50]. Corticosteroid therapy is known to reduce sRAGE in the blood of pediatric patients with ARDS [51]. We revealed that the concentration of sRAGE positively correlated with the plasma level of MCP-1, MIP-1α, and total CT score of MSC-treated patients. In our study, sRAGE slightly decreased after MSC therapy. In contrast, the elevation of sRAGE in the plasma of Control patients on days 14 and 28 compared to the initial day indicated the activation of inflammatory processes in the lungs. In addition, a significant reduction in sRAGE in Control patients coincided with the period of corticosteroid therapy.

Decreasing the cytokine levels in the blood plasma may not reflect the reduction of lung inflammation or the improvement of the lung epithelium. Wick et al. showed that MSC reduces cytokine levels more in lung airspace compared to plasma [52]. IP-10 is a prognostic marker in COVID-19 [53], and its growth in plasma was reported in severe cases [54,55]. MSC transplantations did not affect IP-10 levels in critically ill COVID-19 patients [20,21]. However, in our study, the level of IP-10 decreased after MSC transplantation on day 7, but it remained higher in MSC patients than in Control patients on days 14 and 28. Opposite to findings reporting no changes in MIP-1α plasma level after MSC therapy, as described earlier [41], our findings showed higher levels of MIP-1α on days 7 and 28 in the MSC group. Our results regarding the high level of G-CSF in the MSC group on days 14 and 28 are contrary to the reported association of high levels of G-CSF with COVID-19 severity and the progression of inflammation in COVID-19 patients [56,57] because we did not observe any deterioration in MSC-treated patients. Even though IL-10 has anti-inflammatory properties, some studies show a pro-inflammatory effect in COVID-19 patients [58,59,60]. We observed a significant influx of IL-10 in the MSC group after final transplantation compared to the Control group. Thus, we were the first to show that MSC transplantation in severe COVID-19 patients led to the elevation of plasma levels of pro-inflammatory cytokines such as IP-10, MIP-1α, G-CSF, and IL-10 without disease aggravation.

In the current study, we did not observe any effect of triple MSC transplantation on IL-2 levels. MCP-1 levels increased slightly during the first week but remained unchanged until day 28 in contrast to the Control group, in which it dramatically increased from the second week. Previously, three UC-MSC infusions led to a decrease in MCP-1 levels in a clinical trial [41]. Meta-analysis done by [61] shows that MSC transplantation for COVID-19 has a remarkable effect on efficiency without altering blood levels of CRP, IL-6, and IL-2. Interestingly, transplantation of MSC at stage IIa studies found no significant changes in IL-6, although the indicators of pneumonia on CT improved [20,36,62], which coincides with our results.

The contribution of miRNAs to inflammation is supported by previous data on the upregulated expression of miR-221 in mice with LPS-induced acute lung injury [63], increased levels of miR-133a-3p in atherosclerotic thrombotic cerebral infarction and cardioembolic stroke [64], and higher miR-21 content in COVID-19 patients than in non-COVID-19 volunteers [65] and its upregulation in CD14+ cells among patients with axial spondyloarthritis [66]. Numeral studies indicate miR-146, miR-27, and miR-126 are potential inflammatory biomarkers for ARDS [67,68,69,70,71]. Circulatory miR424-5p is an inflammation marker detected after marathon runs [72]. MiR424-5p is elevated in acute myocardial infarction and autoimmune disease pemphigus [73,74]. In patients with asymptomatic COVID-19, this microRNA is less expressed than in symptomatic COVID-19 [75]. miR-92a-3p is considered an effective biomarker in diagnosing the acute exacerbation of chronic obstructive pulmonary disease, and its levels are elevated in the plasma of patients with COPD [76]. Increased levels of miR-92a-3p are observed in COVID-19, the development of acute lung damage from liposaccharides in animals, and other pathologies of the vascular system [77,78,79]. In our study, the plasma level of pro-inflammatory miR-146a, miR-27a, miR-126, miR-221, miR-21, and miR-133 did not differ between groups and decreased during the observation period. However, after conventional treatment withdrawal, we observed a significant elevation of miR-92a-3p and miR424-5p in patients of both the MSC and Control groups. We also showed that the plasma levels of miR-424-5p and miR-92a-3p negatively correlated with lung lesions. Furthermore, the plasma levels of both miR-92a-3p and miR-424-5p were negatively associated with CRP, SP-D, and IP-10, which, together with the data mentioned above, may indicate their compensatory role in the reduction of inflammation.

The current study demonstrates that UC-MSC increases the expression of genes involved in the development of the immune response, particularly neutrophil activation, leukocyte migration, and phagocytosis, and decrease the expression of apoptosis-associated genes in PBMC from COVID-19 patients. Wharton’s jelly MSC are known to activate neutrophils. In particular, they increase respiratory burst and phagocyte activity and decrease apoptosis [80,81].

At the same time, the fact that UC-MSC reduced the expression of genes involved in proliferation in PBMC indicates it has a certain immunosuppressive effect. Interestingly, we observed an increase in the expression of genes associated with ferroptosis in COVID-19 leukocytes, which may have a positive impact because the increased expression of ferroptosis-related genes was reported in PBMC during the acute phase as well as decreased expression in the recovery phase [82]. Furthermore, it was already reported that the MSC-mediated immunomodulatory effect on PBMC occurs through the activation of NF-kappa B signaling [83]. In addition, we observed a twice-increased level of *PD-L1* expression in PBMC under co-culture conditions (Appendix A). MSC can orchestrate T-cell responses via cell–cell contact interactions and soluble factor secretion. Inhibition of T cell proliferation occurs due to the interaction of PD-1 with its ligand [84]. In addition, soluble factors such as PGE2 and IDO are involved in T cell inactivation, G0/G1 arrest, apoptosis, and inhibition of inflammatory cytokine production [84]. Interestingly, genes involved in the production of PTGES and IDO2 were upregulated in PBMC after being co-cultured with UC-MSC (Appendix A). We show that the conditioned media from UC-MSC and PBMC co-cultures were rich in IL-6, MCP-1, and G-CSF and depleted in MIP-1α and IL-12, which tends toward accordance with the RNA-seq data on the expression levels of the relevant genes in PBMC. Similar data regarding the influence of MSC on the secretion of IL-6, MCP-1, MIP-1α, and IL-12 by PBMC were reported earlier [85,86]. On the other hand, the fact that UC-MSC led to a drop in IL-10 when co-cultivated with PBMC does not agree with the data when PBMC from healthy donors were used. Interestingly, decreased expression of *IL-10* was observed after co-cultivation of PBMC from COVID-19 patients with dental pulp-derived MSC [87]. A putative mechanism of UC-MSC action on immune cells in patients compared to an in vitro model is presented in Figure 7.

Our study shows that UC-MSC triggers the expression of CD69 as an early- and CD25 as a mid-stage T cell marker, as was previously demonstrated [36]. Interestingly, the percentage of effector T cells and effector senescent CD8 and CD 4 T cells decreased under UC-MSC’s influence, corresponding to the CXCR3 expression levels in PBMC seen in RNA-seq. Moreover, the reduction in the pool of effector T cells after co-culturing activated PBMC with UC-MSC was reported previously [88].

In this way, our experimental data reflects the multifunctional response of PBMC to the immunomodulatory effect of UC-MSC. Therefore, our findings show that MSC reduces lung fibrosis while not representing strong immunosuppressive properties; instead, they have an immunomodulating effect. Further research is needed to assess the therapeutic efficacy and mechanisms of UC-MSC infusion in patients with inflammatory airway disorders and ARDS, particularly that caused by COVID-19.

## 4. Materials and Methods

### 4.1. Participants and Study Design

A total of 28 confirmed cases of patients with COVID-19 at the Kyiv City Clinical Hospital #4 were included in this single-center, non-randomized, open-label study registered on clinicaltrials.gov (NCT04461925) [89]. Fifteen patients who had undergone conventional treatment were enrolled in the Control group, and in the MSC group, there were thirteen patients who, in addition to conventional therapy, were treated with three consecutive doses of umbilical cord-derived MSC. Some underlining comorbidities and basic characteristics of both groups are indicated in Table 1. The study protocol was designed following the Declaration of Helsinki and approved by the ethics committee of Kyiv City Clinical Hospital #4 (protocol #280, 23 April 2020). All subjects enrolled in the study signed informed consent statements. All patients met the criteria for moderate COVID-19 severity according to the interim guidelines from the WHO and the Novel Coronavirus Pneumonia Diagnosis and Treatment Plan issued by the National Health Commission of the People’s Republic of China (Provisional 7th Edition) [90]. Patients from both groups had the following conditions: respiratory distress, RR ≥ 30 times/min, and oxygen saturation (SpO2) ≤ 93% at rest, and two-sided pneumonia was observed in all enrolled participants. Patients with severe COVID-19 symptoms persisting after 7–11 days of standard treatment participated in this study. The major treatments for patients included drug therapy, such as antibiotic therapy and standard treatment with dexamethasone to inhibit the inflammatory response. No mechanical ventilation was applied, but patients got oxygen through a facial mask if necessary. Data from each patient were collected for 28 days (Figure 8). Exclusion criteria included the following: shift to other treatment modalities (according to the treating physician), pregnancy, breastfeeding, malignant tumors, other systemic severe diseases, psychosis, enrollment in other clinical trials, co-infection of HIV or syphilis, inability to provide informed consent or to comply with test requirements, invasive ventilation lasting more than 48 h, and the presence of other organ failures (need for organ support).

### 4.2. Respiratory Pathogen Detection

Laboratory validation of SARS-CoV-2 was performed at the Kyiv City Clinical Hospital #4 using real-time polymerase chain reaction (RT-PCR). Briefly, throat swab samples were obtained from the upper respiratory tract and stored immediately in the viral transport medium. After extraction of total RNA, RT-PCR was performed to identify the virus. Genotyping of the SARS-CoV-2 virus was not performed, but the delta strain was the dominant one in Ukraine in that time.

### 4.3. UC-MSC Preparation

Umbilical cords (Ucs) delivered after Caesarean section were collected from 23- to 36-year-old donors at 39–41 weeks of gestation in the Kyiv city maternity hospital #3. All donors (n = 19) provided written informed consent for the sourcing and the usage of their Ucs for the approved clinical study. One week before Uc collection, apparent healthy donors passed the screening for infectious diseases using the serology (anti-HIV1/2, anti-HCV, anti-HBV, anti-Treponema pallidum, and anti-CMV IgG and IgM) and qRT–PCR (presence of nucleic acids of *HIV1/2*, *HCV*, and *HBV*) methods. Using validated PCR kits, UC tissues were tested for *HSV-1/2*, *HHV-6*, *Ureaplasma* spp., and *Mycoplasma genitalium*. Ucs were minced with scissors into small pieces (1–3 mm) and washed intensively on a shaker in Hanks’ balanced salt solution (HBSS) (Sigma, Irvine, UK) supplemented with 100 U/mL penicillin (Arterium, Kyiv, Ukraine) and 50 mg/mL streptomycin (Arterium, Kyiv, Ukraine) until the washing solution became colorless. UC pieces of tissue were plated into cell culture-treated flasks (Sarstedt, Nümbrecht, Germany) and covered with MEM alpha modification (Sigma, Irvine, UK) supplemented with 15% FBS (Sigma, Paraguay origin, Saint Louis, MO, USA), 1× RPMI amino acid solution (Sigma, Irvine, UK), and 1× streptomycin/penicillin (Sigma, Irvine, UK), referred as completed cultural media. Explants were incubated at +37 °C in the presence of 5% CO_2_ for 14 days, with media changed twice a week. When the outgrowth of cells reached 80–90% confluence in a monolayer, they were detached using 0.05% trypsin and 0.02% EDTA (Sigma, Irvine, UK), washed, counted, and passaged at the inoculation density of 4–5 × 10^3^ cells/cm^2^ on culture-treated flasks, referred to as passage 1 (P1). UC-MSC at P3 were harvested and cryopreserved using a rate-controlled freezer at a final concentration of 5% dimethyl sulfoxide (Sigma Aldrich, Saint Louis, MO, USA) in HBSS (Sigma, Irvine, UK). Representative images of the proliferated UC-MSC, surface immunophenotype, differentiation, and karyotype analysis are presented in Appendix A. Aliquots from all samples were collected for quality control. This includes determination of viability by using the trypan blue exclusion method and expression of cell surface markers by using flow cytometry, cytogenetics analysis by using the GTG-banding method, microbiological tests (Bact/Alert 3D, Biomerieux, Durham, NC, USA), and detection of *Mycoplasma* spp. (KIT MycoAlert™ PLUS Mycoplasma Detection, Lonza, Rockland, Miami, FL, USA), according to the manufacturer’s instructions. These quality control tests were performed before each batch of cells was released.

### 4.4. UC-MSC Transplantation

The release criteria for the clinical use of UC-MSC included the absence of contamination with pathogenic microorganisms (bacteria, mycoplasma, and fungi), a normal karyotype, identity, purity patterns characterized as positive (≥95%) for CD73, CD90, and CD105 and negative (≤2%) for CD45, and CD34 expression of cell surface markers according to the minimal criteria for multipotent mesenchymal stromal cells issued by ISCT [91]. During transplantation, we examined and evaluated the frequency and character of every adverse event to clarify if it was connected to the administration of the UC-MSC.

For infusion, UC-MSC at P3 were thawed using a water bath preset at +37 °C until the liquid phase appeared. Cells were centrifuged at 300 g during 5 min RT, and pellets were resuspended in a final volume of 50 mL of vehicle solution composed of saline (Darnytsya, Kyiv, Ukraine) and 5% human serum albumin (Biopharma, Kyiv, Ukraine). The averages of cell viability were 87.8% ± 5.1% before infusion. Cells were infused in three consecutive doses on treatment days 0, 3, and 6 as 1 × 10^6^ cells/kg intravenously. A standard blood transfusion system fitted with a 100 m pore size was used for the infusion. Within 20 min, the UC-MSC were infused dropwise while the patient was monitored electrocardiographically. At the infusion time and through the following 30 min, we continuously checked the patient’s blood pressure, body temperature, pulse, and skin color.

### 4.5. CT Evaluation and Scoring

Following admission, all patients in the supine position were subjected to high-resolution plain chest CT scanning using a Philips Brilliance CT 64 slice scanner (Philips Medical Systems Technologies Ltd., Haifa, Israel), applying a slice thickness of 1 mm with 120 kV and 335 mAs. CT images were analyzed on weeks 2, 8, 24, and 48. The processing and grading of CT images considered such radiologic features as ground glass opacity, reticulation, and honeycombing. The approach applied for the quantitative determination of the affected lung area was as described by Büttner et al. [92] with some changes. Briefly, the affected lung area was measured in polygonal regions of interest in one image at three levels (upper point—above the level of the carina, lower point—below the highest point of the right diaphragm, and middle point—between the previous two, right at the midpoint). Each image was divided into four quadrants and further divided into five sub-quadrants covering 5% of the total image area. The scale applied for evaluation included seven values: 0 (no involvement), 1 (≤10% involvement), 2 (11–20% involvement), 3 (21–30% involvement), 4 (31–40% involvement), 5 (41–50% involvement), 6 (>50% involvement). The total severity score was the sum of the scores of the five lung lobes.

### 4.6. Blood Collection

Blood samples (12–20 mL) of 28 patients with COVID-19 were collected on the day of admission (day 0) and on days 7, 14, and 28 after admission. Briefly, 5 mL was used for routine blood assays completed using a Swelab Alfa Basic hematology analyzer (Boule Medical AB, Spånga, Sweden) at the Kyiv City Clinical Hospital #4. The remaining portions of blood samples were immediately transported to the Institute of Cell Therapy, where the plasma and serum were separated, snap-frozen, and stored at −80 °C for cytokine detection and miRNA analysis.

### 4.7. Cytokine Measurement

The C-reactive protein content in patient sera was determined using AccuBind (Monobind, Lake Forest, CA, USA) according to the manufacturer’s instructions. The detection limit was 0.014 µg/mL. For the detection of G-CSF, IL-2, IL-6, IP-10, MIP-1α, MCP-1, SP-D, sRAGE, and NT-proBNP, an enzyme-linked immunosorbent assay (ELISA) was performed using the Invitrogen kit according to the manufacturer’s instructions. The following ELISA and standard curves were employed to measure each parameter: human G-CSF (BMS2001INST), IL-2 (BMS221INST), IL-6 (BMS213INST), IP-10 (BMS284INST), MIP-1α (KAC2201), IL-10 (BMS215INST), IL-12p70 (KAC1568), TnI (EHTNNI3), and MCP-1 (BMS281INST) from Instant ELISA (Invitrogen, Thermo Fisher Scientific, Vienna, Austria), RAGE (RAB0007) from Sigma (Sigma-Aldrich Chemie GmbH, Steinheim am Albuch, Germany), and SP-D (DSFPD0) and NT-proBNP (DY3604-05) from R&D Systems (Bio-Techne Ltd., Abingdon, UK). The sensitivity levels were 11 pg/mL for G-CSF, 2.3 pg/mL for IL-2, 0.92 pg/mL for IL-6, 1 pg/mL for IP-10, 2 pg/mL for MIP-1α, 0.66 pg/mL for IL-10, 0.2 pg/mL for IL-12p70, 100 pg/mL for TnI, 2.31 pg/mL for MCP-1, 0.11 ng/mL for SP-D, and 3pg/mL for RAGE. The sensitivity level for NT-proBNP was not indicated by the manufacturer. All absorbance measurements were carried out using a HumaReader HS (Human GmBH, Wiesbaden, Germany). All assays were performed in duplicate.

### 4.8. miRNA Expression

miRNA was extracted from the plasma of patients with COVID-19 and of the age-matched control group according to the instructions for the NucleoSpin miRNA Kit (Macherey-Nagel, Hoerdt, France) and stored at −80 °C. The concentration of isolated miRNA was measured using a NanoDrop 2000 spectrophotometer (Thermo Scientific Inc., Wilmington, DE, USA), and miRNA was reverse transcribed into cDNA using the miRNA 1st-Strand cDNA Synthesis Kit (Agilent Technologies, Lexington, MA, USA) with a universal reverse primer from the synthesis kit. Quantitative RT-PCR was conducted to detect miRNA levels using a 5× HOT FIREPolEvaGreen qPCR Mix Plus kit (no ROX) (Solis BioDyne, Tartu, Estonia) with a CFX96™ Real-Time PCR Detection System (BIO-RAD Laboratories, Inc., Singapore). For each sample, the qRT-PCR reaction consisting of a 15 min hot start at 95 °C for polymerase activation, followed by 44 cycles of 15 s at 95 °C and 20 s at 60 °C, was performed in triplicate. The 2^ΔΔCq^ method [93] was used for miRNA quantification analysis, with U6 as a reference. The primer sequences are listed in Appendix A.

### 4.9. Isolation of PBMC and Co-Culture with UC-MSC

Human PBMC were isolated via separation with Ficol (Cytiva, Global Life Sciences Solutions, Marlborough, MA, USA) from peripheral blood samples of patients with ARDS caused by COVID-19. All PBMC (n = 6) were cryopreserved with 10% DMSO (Sigma, Saint Louis, MO, USA) in FBS (Sigma, Paraguay origin, Saint Louis, MO, USA) and stored in liquid nitrogen until processed. UC-MSC from three donors were thawed and cultured as described above. The cells were incubated with 20 µg/ml mitomycin C in completed cultural media for 2 h, detached using 0.05% trypsin and 0.02% EDTA (Sigma, UK), washed, counted, mixed in equal proportion (1:1:1), and passaged at the inoculation density of 1 × 10^5^ per well of 24-well plate. PBMC were thawed in a water bath preset at +37 °C until the liquid phase appeared. Cells were centrifuged at 300× *g* during 10 min RT, and pellets were resuspended in an RPMI-1640 (Gibco, Life Technologies Corp., Carlsbad, CA, USA) with 10% heat-inactivated FBS. PBMC (5 × 10^5^) were placed into 0.4 µmThinCerts™-TC insert (Greiner Bio-One, Monroe, NC, USA) in a total volume of 500 μL in the presence of Dynabeads^®^ Human T-Activator CD3/CD28 (Life Technologies AS, Oslo, Norway) at a bead/cell ratio of 1:5. Cells were incubated at 37 °C over 6 days and collected for FACS and RNAseq analyses. The conditioned media from UC-MSC/PBMC were collected for further ELISA analysis.

### 4.10. RNA Isolation, RNA-seq, and Bioinformatics Analysis

The samples were prepared in biological triplicate. RNA was extracted using a Nucleospin RNA isolation kit (Macherey–Nagel, Hœrdt, France) according to the manufacturer’s protocol. RNA-seq libraries were prepared using the Agilent SureSelect Automated Strand-Specific RNA Library Prep, with polyA selection by Novogene Co., LTD (Beijing, China). Prepared libraries were sequenced on an Illumina HiSeq2000, utilizing a paired-end 150 bp sequencing strategy (short-reads) and 20M read pairs per sample. Raw data (raw reads) of the fastq format were first processed through fastp software. In this step, clean data (clean reads) were obtained by removing reads containing adapters, reads containing poly-N, and low-quality reads from raw data. At the same time, the Q20, Q30, and GC content of the clean data were calculated. All of the downstream analyses were based on clean data with high quality. Raw paired-end sequence reads were mapped to the human transcriptome (ensembl_homo_sapiens_grch38_p12_gca_000001405_27) using Hisat2 v2.0.5. featureCounts v1.5.0-p3 was used to count the read numbers mapped to each gene. Then, the FPKM of each gene was calculated based on the length of the gene and the read count mapped to this gene. Differential expression analysis was performed using the DESeq2 R package (1.20.0). Genes with adjusted *p*-value < 0.05 and |log2 (FoldChange)| > 0 were considered differentially expressed. Gene Ontology (GO) enrichment analysis of differentially expressed genes and the statistical enrichment of differential expression genes in KEGG pathways were implemented by using the clusterProfiler R package. The differentially expressed genes are listed in Appendix A. Clean data were deposited in the NCBI Sequence Read Archive and can be accessed under PRJNA929329. The local version of the GSEA analysis tool (http://www.broadinstitute.org/gsea/index.jsp (accessed on 4 January 2023)) was used for Gene Set Enrichment Analysis (GSEA). GO and the KEGG data set were used for GSEA independently.

### 4.11. Statistical Analysis

SPSS version 27.0 software (IBM Corp. Released 2020. IBM SPSS Statistics for Windows, Version 27.0. Armonk, NY, USA: IBM Corp.) was used for statistical analysis. GraphPad Prism software (version 7.0a, Inc. San Diego, CA, USA) was used for data visualization. The variables were presented as medians with interquartile ranges. The Wilcoxon signed-rank test was used to compare the time-dependent events. The Mann–Whitney U test was used to compare the differences between the groups at each time point. The paired *t*-test was applied to compare related groups. The Spearman rank test was performed to test the correlations between variables. Statistical significance was set at a two-tailed *p*-value of ≤0.05.

## 5. Conclusions

Triple MSC transplantation in patients with moderate/severe COVID-19 was shown to be safe and to improve lung lesions over a one-year follow-up period. Although the majority of parameters did not show a significant difference between the MSC group and the Control group after treatment, combined data still indicate cell therapy tends to reduce inflammation and benefit the patients. MSC transplantation leads to a decrease in markers of inflammation in patients (ESR, CRP, and banded neutrophils), more rapid recovery of blood lymphocytes, and reduced SP-D as one of the main markers of lung injuries. On the other hand, the increased plasma levels of some pro-inflammatory cytokines such as IP-10, MIP-1α, G-CSF, and IL-10 could suggest a more immunomodulatory effect of MSC rather than immunosuppression in patients with ARDS caused COVID-19. In vitro, UC-MSC demonstrated an immunomodulatory effect on PBMC, namely, an increase in the activation of neutrophils, phagocytosis and migration of leukocytes, and activation of early markers of T cells, and a decrease in the maturation of effector and senescent effector T cells.

## Figures and Tables

**Figure 1 ijms-24-04435-f001:**
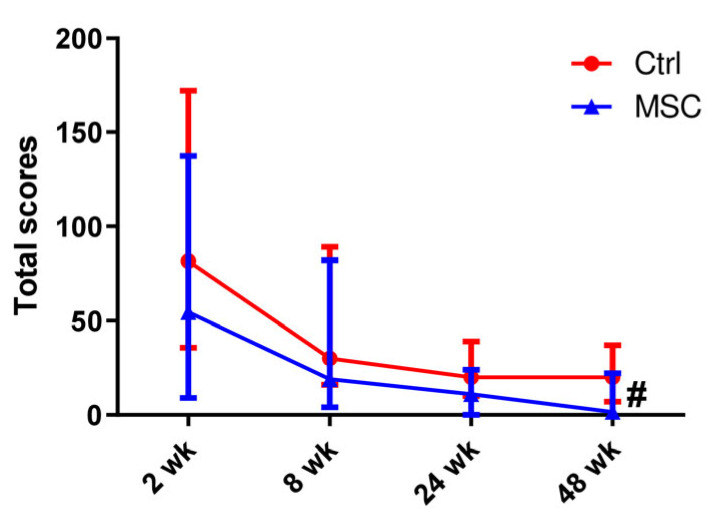
Dynamic changes in lung CT scores in patients with COVID-19. Data are presented as a median and IQR. Mann–Whitney U test for comparing the control and MSC groups at each time-point: #, *p* ≤ 0.05. The red line represents the Control group; the blue line represents the MSC group.

**Figure 2 ijms-24-04435-f002:**
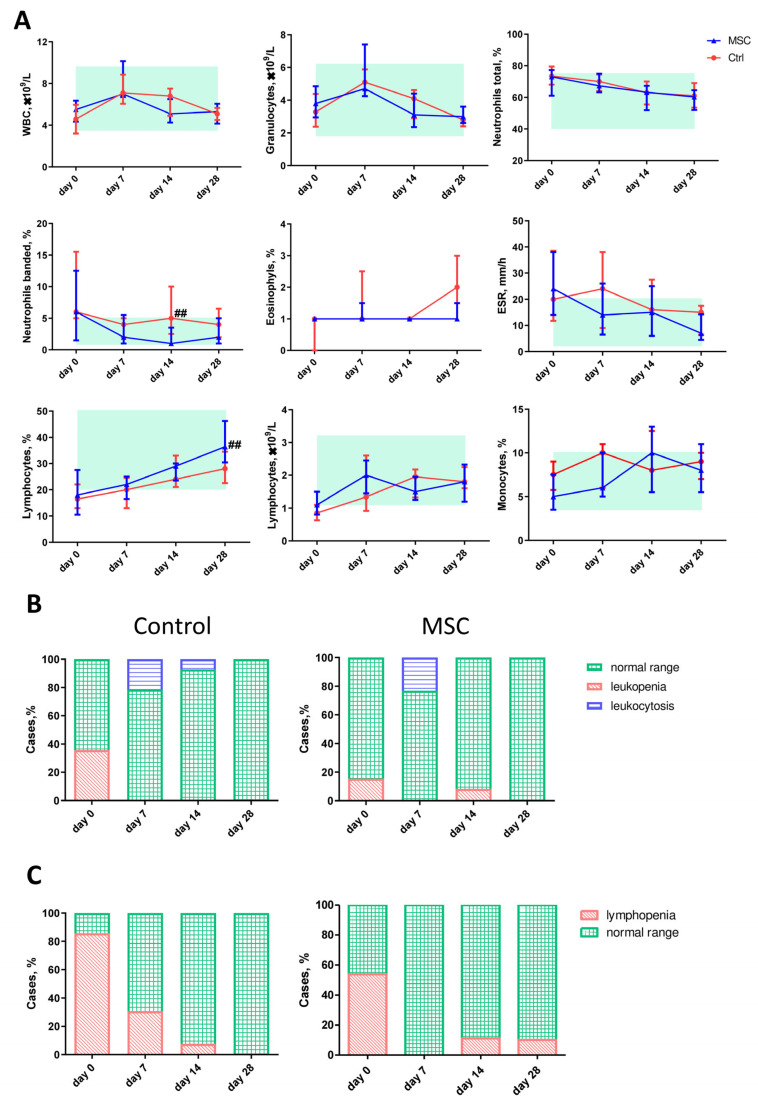
Dynamic changes in hematological parameters in patients with COVID-19. (**A**) The dynamic profile of blood cell types and ESR. A light green rectangle denotes the normal range. (**B**) The ratio of leukopenia and leukocytosis. (**C**) The ratio of lymphopenia. Data are presented as the median and interquartile range. Mann–Whitney U test for comparing the control and MSC groups at each time-point: ##, *p* ≤ 0.01. WBC, white blood cells; ESR, erythrocyte sedimentation rate. The red line represents the Control group; the blue line represents the MSC group.

**Figure 3 ijms-24-04435-f003:**
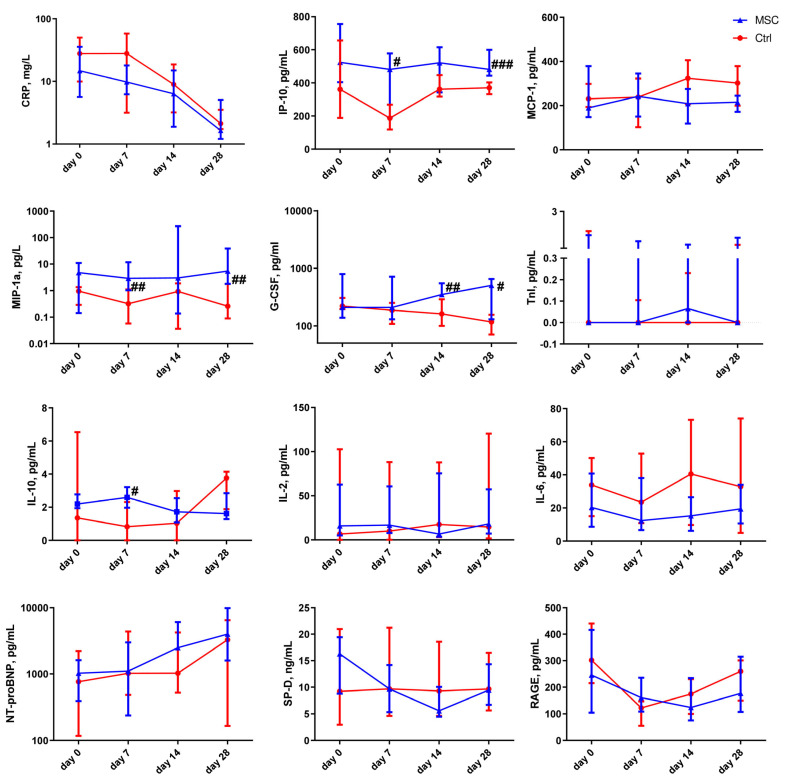
Dynamic changes in cytokine levels in patients with COVID-19. Data are presented as a median and interquartile range. Mann–Whitney U test for comparing the control and MSC groups at each time-point: #, *p* ≤ 0.05; ##, *p* ≤ 0.01; ###, *p* ≤ 0.001. The red line represents the Control group; the blue line represents the MSC group. CRP, C-reactive protein; IP-10, interferon gamma-induced protein 10; MCP-1, monocyte chemoattractant protein-1; MIP-1α, macrophage inflammatory protein-1 alpha; G-CSF, granulocyte colony-stimulating factor; IL-10, interleukin 10; IL-2, interleukin 2; IL-6, interleukin 6; NT-proBNP, N-terminal prohormone of brain natriuretic peptide; TnI, troponin I; SP-D, surfactant D; RAGE, receptor for advanced glycation endproducts.

**Figure 4 ijms-24-04435-f004:**
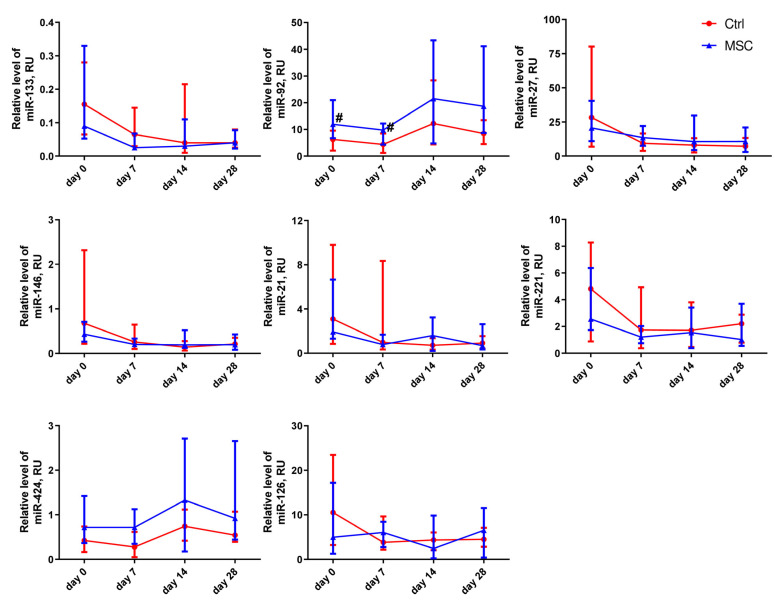
Dynamic changes in miRNAs’ expression levels in patients with COVID-19. Data are presented as a median and IQR. Mann–Whitney U test for comparing the control and MSC groups at each time-point: #, *p* ≤ 0.05. The red line represents the Control group; the blue line represents the MSC group.

**Figure 5 ijms-24-04435-f005:**
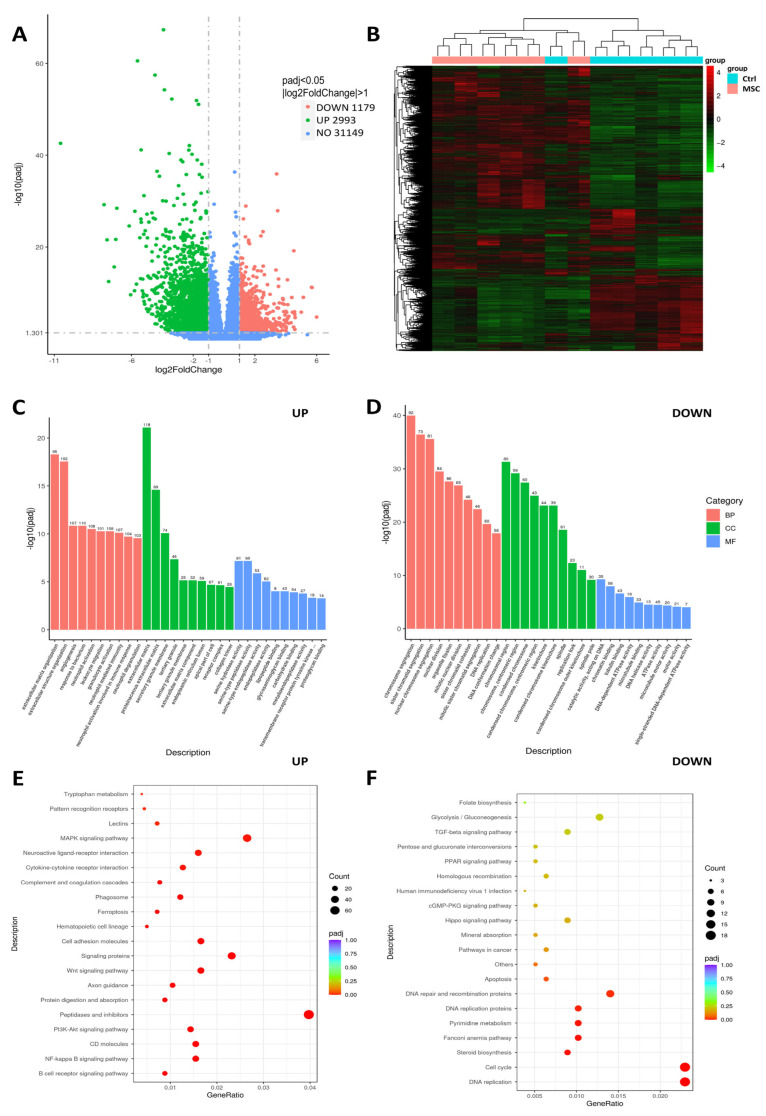
Transcriptome profile of COVID-19 patients-derived PBMC cultured with/without presence of UC-MSC. (**A**) Volcano plot for differentially expressed genes. (**B**) Heatmap represents numbers of differentially expressed genes between groups. (**C**,**D**) Gene Ontology representation analysis of the 2063 upregulated and 1433 downregulated genes. (**E**,**F**) KEGG annotation of differentially expressed genes. Experiments were performed in biological duplicate. The genes with padj < 0.05 were considered significant; log2Fold < 1, downregulated; log2Fold > 1, upregulated.

**Figure 6 ijms-24-04435-f006:**
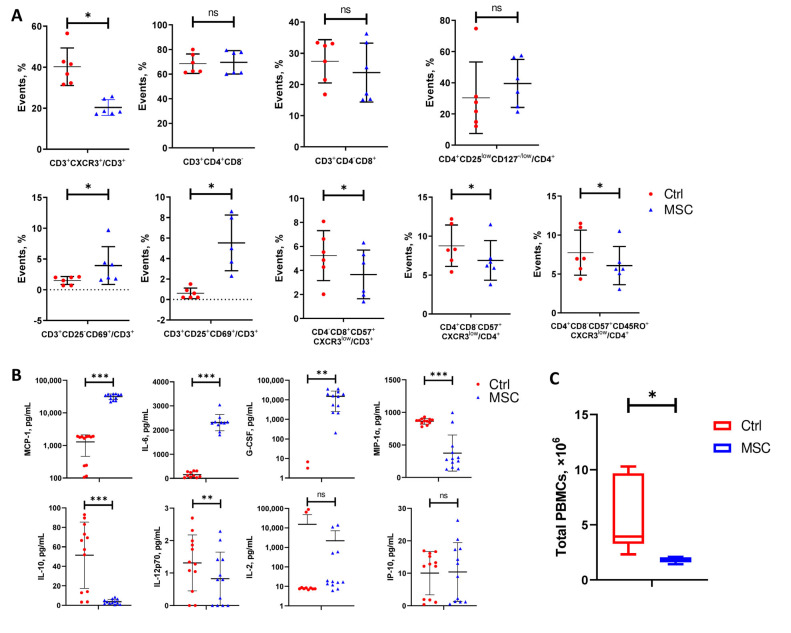
Changes in cell subpopulations’ ratios of COVID-19 patients-derived PBMC, cytokine levels in media, and total PBMC count after 6 days of co-culturing with or without UC-MSC. (**A**) Frequency of different T cells’ subpopulations. (**B**) Cytokine levels. (**C**) Difference between groups in total PBMC count collected from 6/24 wells. Data are presented as the mean with SD. Paired t-test was used for comparison between groups: *, *p* < 0.05; **, *p* ≤ 0.01; ***, *p* ≤ 0.001; ns, not significant. The red circle represents the Control PBMC group; the blue triangle represents the PBMC+UC-MSC group.

**Figure 7 ijms-24-04435-f007:**
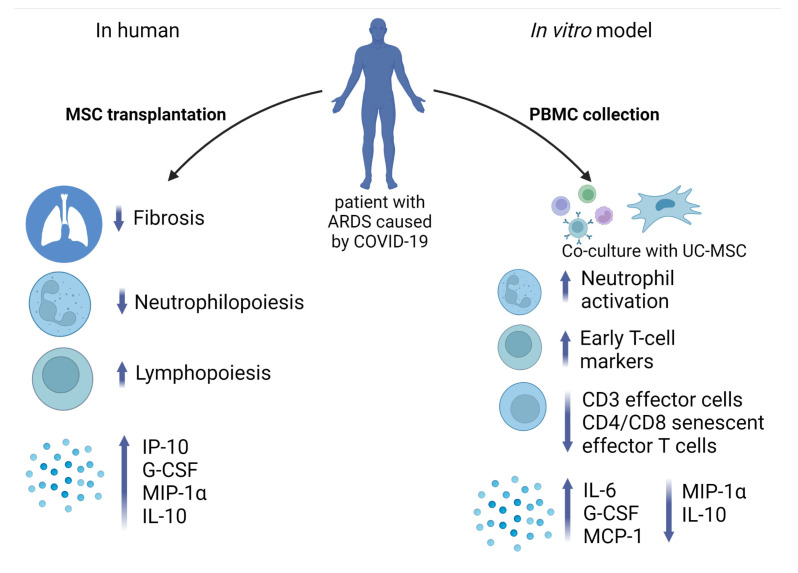
Immunomodulation effect of UC-MSC in vivo and in vitro. UC-MSC after transplantation in patients with ARDS caused by COVID-19 led to the reduction of lung tissue fibrosis and neutrophil count, stimulating lymphopoiesis and production of proinflammatory cytokines. In vitro model of PBMC co-culture with UC-MSC revealed abundance in neutrophil activation, early T-cell marker expression, and cytokine production, whereas counts of effector T cells and senescent effector CD4 and CD8 cells declined. Created with BioRender.com (accessed on 14 February 2023).

**Figure 8 ijms-24-04435-f008:**
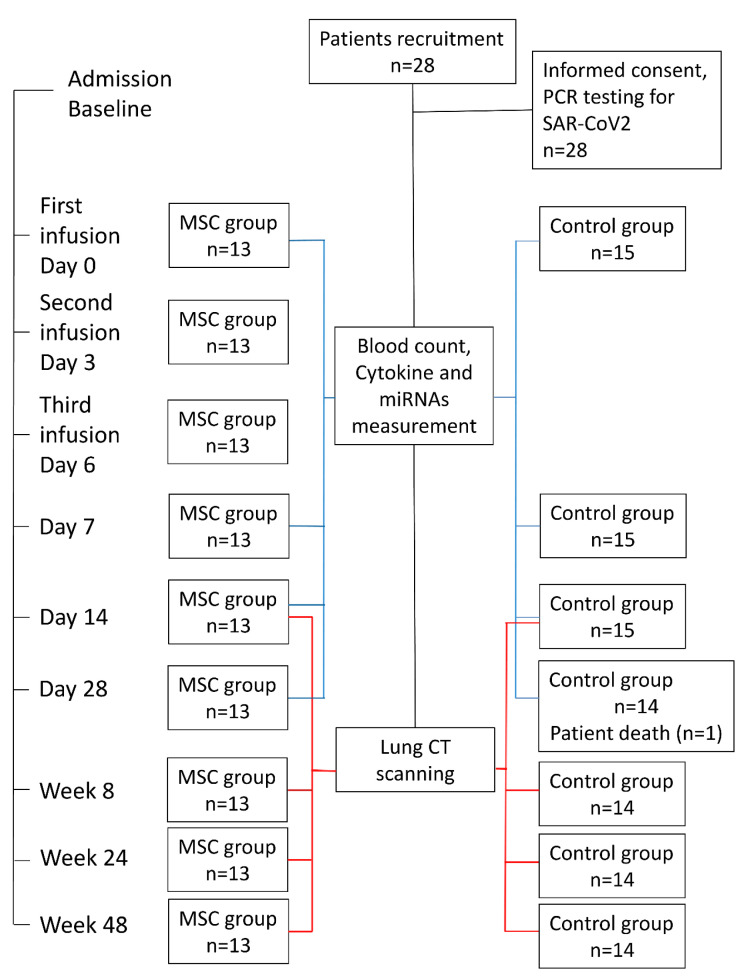
Flow chart for patient enrollment, intervention, and follow-up.

**Table 1 ijms-24-04435-t001:** Clinical characteristics of patients with COVID-19 included in this study.

Parameter	Control Group	MSC Group	*p*-Value
	(n = 15)	(n = 13)	
Age, years (median, range)	62 (32–73)	58 (32–71)	0.25
Gender			0.61
Male	11/15 (73.33%)	8/13 (61.5%)	
Female	4/15 (26.67%)	5/13 (38.5%)	
The interval from illness onset to hospital admission (days)	11.0 (8.5–12.8)	12.3 (9–14.5)	0.29
Underlying diseases—no. (%)			
Hypertension	9/15 (60.0%)	5/13 (38.5%)	0.34
Diabetes	1/15 (6.67%)	1/13 (7.7%)	0.91
Heart disease	6/15 (40.0%)	4/13 (30.7%)	0.68
Symptoms—no. (%)			
Fever	11/15 (73.33%)	11/13 (84.6%)	0.61
Cough	15/15 (100.0%)	13/13 (100%)	1
Shortness of breath	12/15 (80.0%)	9/13 (69.2%)	0.65
Diarrhea	1/15 (6.67%)	0/14 (0%)	0.78
Fatigue	15/15 (100.0%)	13/13 (100%)	1
Myalgia	3/15 (20.0%)	6/13 (46.1%)	0.25
Clinical outcome—no. (%)			
Recovered and discharged	14/15 (93.33%)	13/13 (100%)	0.78
Death	1/15 (6.67%)	0/13 (0%)	0.79

## Data Availability

All datasets presented in this study are cited in the article.

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
