# Peer review of "Treatment of Acute Respiratory Distress Syndrome Caused by COVID-19 with Human Umbilical Cord Mesenchymal Stem Cells"

_ijms, 2023, doi:10.3390/ijms24054435_

Round 1

Reviewer 1 Report

The study by Bukreieva et al., titled “Treatment of acute respiratory distress syndrome caused by COVID-19 with human umbilical cord mesenchymal stem cells” is well designed and executed. This study aims to characterize the effects of MSC transplantation in COVID19-induced ARDS. The authors have looked at a wide variety of readouts to evaluate the impact of MSCs on COVID-mediated pathogenesis and immune response. Parameters such as CBC, profiling of cytokines that are implicated in inflammation and thrombosis in addition to evaluating mi-RNA expression that regulate COVID pathogenesis and immune response are recorded. The data is presented clearly but the feeble phenotypic differences seen across the board makes it hard to make a single overall conclusion. 

Even though the overall concept of using MSCs for treatment of severe COVID-19 is not novel, the study adds incremental data to the already existing body of work to further demonstrate the approach. The authors do a fair job in discussing the discrepancies that they have observed from what has been previously reported by other groups. Overall, the study needs to be presented in a way that doesn’t leave the readers wondering what the overall conclusion is. In addition to just reporting the results and discussing their findings, it will help the study if the authors illustrate a model to explain how MSCs exert their therapeutic impact.  More importantly, elaborate on the mechanism of action behind the efficacy of the MSC – where do these cells home to? What are the effector cells and how do MSCs shape the inflammatory milieu in a COVID19 infected lung and alter the systemic outcome? In other words, how exactly do MSCs bring about immunosuppression and reduce the immunopathology that’s observed in the form of a cytokine storm? Answering these questions using the data presented should help tie the overall manuscript together and improve the scientific soundness of the manuscript.

Reviewer 2 Report

Dear Authors,

The manuscript entitled " Treatment of acute respiratory distress syndrome caused by 3 COVID-19 with human umbilical cord mesenchymal stem cells" represents an interesting study. However, i have major concerns, before the manuscript will be further processed.

1) In section 4. Materials and Methods-4.1 Participants and study design. I do believe that the number of patients is low. The study includes 28 patients, which means for each group at least are 14 patients. In my opinion, in order to have safe conclusions, regarding this important aspect of the current manuscript at least a number of 25 patients should be enrolled in each group.

2) 4.3 UC-MSC preparation. The authors stated that used MSCs from two donors right? Also, this number of donors is also low. For a kind of this study, a minimum number of 10 different MSCs donors should be included. MSCs are a quite heterogenous population and differences between donors have also been reported previously in literature.

Moreover, the authors should include in their study, images of the proliferated MSCs, flow cytometric, and trilineage differentiation results and cytogenetics analysis results in order to support that the applied cells are belonging to MSCs according to the criteria outlined by the ISCT.

3) In results section 2.3 Computed tomography of the lung and 2.4. The dynamic profile of hematological parameters in in Figures 1 and 2 only the last diagram which represents a combination of the former diagrams is required to be presented. The other diagrams can be included as supplementary data.

4) In the results section 2.5 The dynamic changes in cytokine and miRNA levels in the plasma of patients with COVID-19. The authors quantified certain cytokines. Why the authors concluded to this set of cytokines to be measured. Also in figure 3, the same as above, only the combined diagram is required to be presented.

5) In the results section 2.5 The dynamic changes in cytokine and miRNA levels in the plasma of patients with COVID-19. What was the purpose behind of determining only these specific miRNAs. Also, the authors must perform grouping of the miRNAs and differential expression between the groups. In addition, the authors should detect this set of miRNAs if they are expressed from the MSCs under in vitro culturing conditions.

6) To be honest and checking your results, the patients in the MSCs group did not have any significant change to their parameters compared to the control group (evidence from results) The MSCs may help the patients to combat a little better COVID-19. This should be discussed thoroughly in the discussion section.

7) Please include in your manuscript the following publication https://doi.org/10.37349/ei.2021.00010

8) Please discuss and compare your results with other previously conducted publications in the field such as doi:10.4252/wjsc.v12.i8.731. and doi: 10.4331/wjbc.v13.i2.47.

Round 2

Reviewer 2 Report

Dear Athors,

The majority of my comments have beeen weel addressed by your side. 

However i insist on changing the study from clinical trial to pilot study, given though the small number of patients.